# Conformity Dynamics in Multi-Agent Systems: A Network Topology Perspective

## Abstract

Large language models (LLMs) are increasingly employed as agents in multi-agent systems (MAS), where collective decision-making is shaped by conformity dynamics—the tendency of agents to align their judgments with prevailing majority opinions. Although conformity can mitigate individual noise, it also risks inducing information cascades that culminate in confident yet erroneous consensus. This paper presents the first systematic study of how network topology modulates the strength, speed, and reliability of conformity in LLM-based MAS for misinformation detection. We propose a confidence-normalized pooling framework that balances individual judgment against social influence. Our evaluation contrasts two canonical decision modes: Centralized Aggregation, represented by star and hierarchical networks with single-round hub decisions, and Distributed Consensus, characterized by iterative convergence in network structures ranging from sparse rings to fully connected graphs. Results show that in Centralized topologies, the reliability of collective outcomes is tightly coupled to the competence of the hub agent. In Distributed topologies, greater connectivity and stronger social weighting accelerate convergence and enhance accuracy. Our analysis further underscores the double-edged nature of conformity: while MAS hold promise for misinformation detection, conformity can also drive them into "wrong-but-sure" consensus states, where misclassified claims are sustained with high collective confidence. The code and dataset are available at: `Topology-of-Multi-Agent-Systems`.

## 1 Introduction

Large language models (LLMs) are increasingly deployed as agents that interact and make collective decisions in multi-agent systems (MAS)(Li et al., 2024a; Wei et al., 2025; Zheng & Tang, 2025). Across applications such as misinformation detection(Li et al., 2025), safety auditing(Song et al., 2024), and collaborative problem solving(Du et al., 2024), group performance depends not only on the competence of individual agents but also on the social dynamics shaped by interaction rules and communication structures(Li et al., 2024b). A central mechanism in these dynamics is conformity, whereby agents adjust their judgments toward majority opinions. Prior study indicates that such alignment can promote coordination and suppress individual-level noise(Choi et al., 2025). At the same time, conformity can also trigger self-reinforcing information cascades, causing groups to converge on erroneous conclusions with high confidence(Zhu et al., 2025).

Prior work on MAS with LLM agents has largely emphasized task effectiveness, including the design of debate protocols, the orchestration of specialized roles, and the optimization of coordination efficiency(Agashe et al., 2025; Chen et al., 2025; Grötschla et al., 2025). Recent studies demonstrate that the collective behavior of interacting LLMs is not merely the aggregation of independent votes but is instead shaped by influence dynamics(Ghoshal et al., 2025; Weng et al., 2025). For example, multi-agent debate frameworks reveal that repeated exposure to peer reasoning can shift individual judgments toward majority positions—sometimes enhancing reliability but at other times reinforcing systematic biases(Han et al., 2025). Complementary research on persuasion and deliberation further shows that LLM agents frequently imitate stylistic or argumentative patterns that dominate early in the dialogue, suggesting that conformity pressures can emerge even in the absence of explicit majority signals(Argyle et al., 2025; Salvi et al., 2025).

Conformity has also been studied in computational opinion dynamics and social simulation(Farjam & Loxbo, 2024; Zhou et al., 2025). Classical models such as the DeGroot model or bounded-

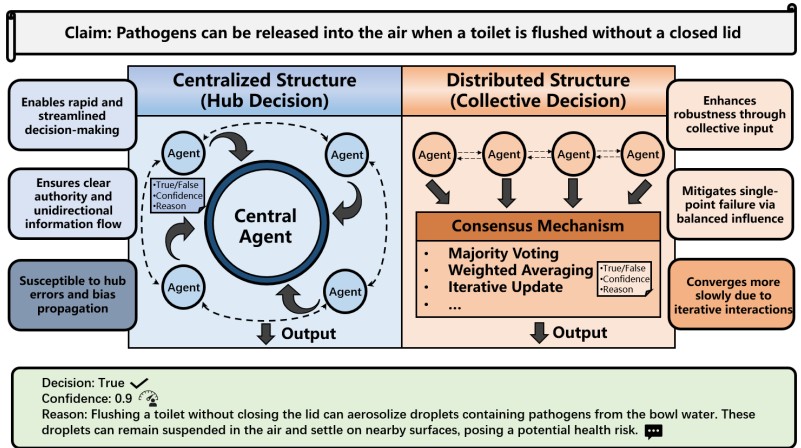

Figure 1: Illustration of Centralized (Hub Decision) and Distributed (Collective Decision) multi-agent structures for claim verification, highlighting the mechanisms, advantages, and limitations.

confidence dynamics formalize how individuals iteratively adjust beliefs based on neighbors, providing theoretical insights into convergence, polarization, and cascades(Helfmann et al., 2023; Li & Porter, 2023). These models have been applied to LLM collectives to probe robustness and diversity of reasoning(Ding et al., 2025). However, most such work treats conformity at an abstract level, without grounding it in the actual mechanisms by which agents generate judgments in concrete decision-making tasks. As a result, while we know that network connectivity and update rules affect consensus, we lack a systematic account of how conformity unfolds in LLM-driven MAS.

This paper addresses this issue by investigating how network topology influences conformity in LLM-based MAS. We formulate group fact-checking as a binary misinformation detection task and propose a update rule that balances each agent's own judgment with the its neighbors through a global confidence parameter. Specifically, we contrast two decision modes: (i) Centralized Aggregation (e.g., star or hierarchical networks), where the hub produces a collective decision in a single round; and (ii) Distributed Consensus (ranging from sparse rings to fully connected graphs), where influence propagates iteratively across neighborhoods. This framing enables us to disentangle one-step hub dominance from multi-round neighbor interaction and to quantify how each induces conformity in the misinformation detection task. Our contributions are summarized as follows:

- **Topology as a determinant of conformity.** In the context of a binary misinformation detection task, we demonstrate that network structure governs the strength, speed, and reliability of collective judgments in LLM-based multi-agent systems, contrasting Centralized Aggregation with Distributed Consensus.

- **A transparent confidence-normalized pooling rule.** We propose a general update mechanism that exposes a single global confidence parameter $\alpha$ to balance self-reliance and social influence, while integrating agent-level confidence to guarantee bounded scores, stable binarization, and reproducible conformity dynamics.

- **Empirical insights on misinformation robustness and risks.** Experiments on the latest fact-checking datasets show that hub competence and majority composition are decisive for system reliability. We further reveal the double-edged nature of conformity: it amplifies accuracy when decisions are reliable, but can also induce "wrong-but-sure" consensus.

## 2 RELATED WORKS

**Multi-Agent Systems and Collective Decision-Making.** MAS provides a general framework for modeling distributed intelligence, where autonomous agents interact, share information, and make collective decisions(Wang et al., 2022). Classic studies in MAS have examined topics such as task allocation, consensus protocols, and communication efficiency(Bao et al., 2022; Amirkhani & Barshooi, 2022), assuming simple agent models or well-defined coordination objectives. The integration of LLMs into MAS has opened new opportunities, enabling agents to deliberate, argue, and reason over complex tasks in natural language, with applications ranging from automated fact-

checking(Han et al., 2025) to policy analysis(Chudziak & Wawer, 2025). However, most studies emphasize task performance metrics, while paying less attention to emergent social dynamics such as conformity, which may crucially shape collective reliability.

**Conformity and Opinion Dynamics.** The phenomenon of conformity has long been studied in social psychology(Gestefeld & Lorenz, 2023), most famously through Asch's experiments showing that individuals often adopt majority views even when they contradict clear evidence(Capuano & Chekroun, 2024). Computational models of opinion dynamics, such as the DeGroot averaging model(Dong et al., 2024) and bounded-confidence dynamics(Li & Porter, 2023), provide formal tools to capture how local influence aggregates into consensus. These models have been applied across sociology and computer science to explain phenomena such as information cascades and polarization. More recently, they have inspired studies of algorithmic collectives(Chuang et al., 2024b;a), where artificial agents update beliefs through similar rules. Yet, such frameworks generally treat conformity at a stylized level, abstracting away from the mechanisms of modern LLM agents, such as confidence calibration, justification generation, and context sensitivity. As a result, the connection between psychological conformity theories, formal opinion models, and concrete decision-making tasks in MAS remains underdeveloped.

**Network Topology and Collective Behavior.** Network science has established that the structure of connections critically shapes diffusion, coordination, and consensus(Cheng et al., 2021; Amirkhani & Barshooi, 2022). **Centralized Aggregation** topologies (e.g., star or hierarchical networks) facilitate rapid alignment but risk single-point failure when the hub is biased or unreliable. By contrast, **Distributed Consensus** topologies (e.g., ring or complete graphs) diffuse influence more evenly, offering robustness at the cost of slower convergence. In the field of MAS, some studies have investigated how connectivity impacts consensus speed, communication cost, or robustness to noise(Li et al., 2024b; Da et al., 2025; Wang et al., 2025; Yang et al., 2025), but the explicit link between network topology and the strength of conformity dynamics remains underexplored. It remains unclear how different network topologies interact with LLM agents' confidence weighting in realistic decision contexts. In this study, we address this gap by comparing **Centralized Aggregation** and **Distributed Consensus** structures in controlled experiments, thereby unifying perspectives from artificial intelligence, social psychology, and network science.

## 3 METHODOLOGY

### 3.1 AGENT DECISION MODEL

Inspired by classical opinion dynamics and weighted consensus protocols(Gu & Tang, 2005; Li & Porter, 2023; Dong et al., 2024), we formalize a binary misinformation detection task for MAS. At each round $t$, agent $i$ evaluates a claim and outputs a binary judgment $y_i^{(t)} \in \{0, 1\}$ (0 =True, 1 =False) together with a confidence score $p_i^{(t)} \in [0, 1]$ indicating its self-assessed reliability.

**Update rule.** Agents revise an internal support score using confidence-normalized pooling:

$$s_i^{(t+1)} = \frac{\alpha \, p_i^{(t)} y_i^{(t)} \; + \; (1 - \alpha) \sum_{j \in N_i} p_j^{(t)} y_j^{(t)}}{\alpha \, p_i^{(t)} \; + \; (1 - \alpha) \sum_{j \in N_i} p_j^{(t)} + \varepsilon}, \tag{1}$$

where $N_i$ is the neighbor set of agent $i$, and $\varepsilon$ ensures numerical stability. Two quantities govern the dynamics: $\alpha \in [0, 1]$ is a global self–social weight trading off independence and peer influence, while $p_i^{(t)}$ is an agent-specific confidence that scales both the persistence of its own judgment and the influence exerted on neighbors. Thus, even under strong conformity (small $\alpha$), a large $p_i^{(t)}$ can preserve an agent's stance; conversely, even under high self-reliance (large $\alpha$), strong high-confidence neighbor signals can still sway it. By construction, $s_i^{(t+1)} \in (0, 1)$.

**Binary readout.** The score is mapped to a binary label via a fixed threshold $\tau$:

$$y_i^{(t+1)} \; = \; \mathbb{1}\big[s_i^{(t+1)} \geq \tau\big]. \tag{2}$$

We adopt $\tau = 0.5$ by default, so $s_i^{(t+1)} < 0.5$ yields True (0) and $s_i^{(t+1)} \geq 0.5$ yields False (1), preserving class symmetry and ensuring a stable, interpretable binarization across tasks and confidence distributions.

## 3.2 PROMPT DESIGN

For each agent, we design a standardized prompt to generate decisions in a reproducible manner as shown in Figure 2. It consists of three components that jointly guide the agent's reasoning process. First, we provide a concise background profile, automatically generated by LLMs, which situates the claim within its relevant domain context and contains three to four sentences. Second, we include a task description that explicitly instructs the agent to decide whether the claim is true or false by relying on the background profile and logical reasoning. Third, we specify the output requirements, which mandate that each agent produce (i) a binary label $y_i^{(t)} \in \{0, 1\}$, (ii) a confidence score $p_i \in [0, 1]$ reflecting its certainty, and (iii) a brief justification (less than 100 words) explaining its reasoning. During the interaction, the three outputs are exchanged among agents and subsequently aggregated. Complete prompts for different networks and agents are provided in Appendix A.

---

**Example Prompts**

**Claim:** Pathogens can be released into the air when a toilet is flushed without a closed lid.

- - - - - - - - - - - - - - - - - - - - - - - - - - - - - - - - - - - - - - - - - - - - - - -

**Background Profile:** You are an expert in public health specializing in how everyday human activities influence the transmission of infectious diseases...
**Task Description:** Carefully read the claim above and evaluate its veracity. Your task is to:
1. Decide whether the claim is True or False.
2. Provide a confidence score p in [0,1] reflecting how certain you are.
3. Give a justification (less than 100 words) explaining the reasoning behind your judgment.
4. Return your answer strictly in the JSON format.

---

Figure 2: Example Prompt for the Misinformation Detection Task.

## 3.3 NETWORK TOPOLOGIES

We investigate how conformity dynamics differ under two canonical modes of decision-making: **Centralized Aggregation** and **Distributed Consensus**. Both modes apply the same update rule in Eq. (1), but their structural properties and evaluation metrics highlight different consensus mechanisms. All structures are instantiated with seven agents. In Centralized Aggregation structures, conformity is immediate and dependent on the hub's confidence and correctness, whereas in Distributed Consensus structures, conformity emerges gradually through interactions. By contrasting one-step hub dominance with multi-round neighbor accumulation, we can distinguish the structural vulnerabilities of centralization from the collective robustness of distributed consensus.

### 3.3.1 CENTRALIZED AGGREGATION

**Centralized Aggregation** structures are characterized by information flowing upward from peripheral or lower-level agents toward a central node, whose judgment ultimately determines the collective outcome. We study two representative forms of such structures as shown in Figure 3:

**(a) Star network:** six peripheral nodes transmit their judgments to a central node, without interacting with one another.

**(b) Hierarchical structure:** agents are organized in three layers with a total of seven agents, where judgments from leaf nodes are first aggregated by intermediate nodes before being passed upward to the root. For consistency, other structures are also instantiated with seven agents.

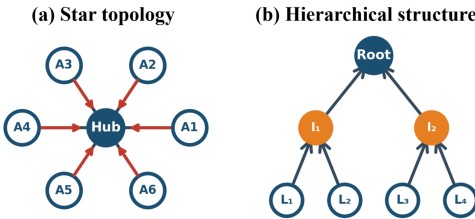

Figure 3: Centralized Aggregation Topology.

**Protocol.** In both structures, decision-making proceeds through a single update round. Peripheral or lower-level agents first submit their judgments to the hub or root, which then integrates this information into its own final decision. The central node's output is directly adopted as the collective decision of the group.

**Metrics.** We evaluate reliability and conformity using three complementary measures: **(1) Central Accuracy (CA):** correctness of the hub or root node's final judgment. **(2) Peripheral Accuracy (PA):** average correctness of all other peripheral nodes. **(3) Center–Periphery Consistency (CPC):** proportion of peripheral nodes whose judgments agree with the central node, quantifying the strength of conformity.

### 3.3.2 Distributed Consensus

**Distributed Consensus** structures distribute influence symmetrically across agents, such that no single node dominates the outcome. We examine a family of ring-to-complete networks by progressively increasing each agent's neighbor count as shown in Figure 4:

**(a) Sparse ring (2 neighbors)**: each agent interacts only with its immediate predecessor and successor.

**(b) Expanded rings (3–5 neighbors)**: each agent connects to a wider local neighborhood, allowing influence to spread more rapidly.

**(c) Fully connected graph (6 neighbors)**: each agent is connected to all others, representing the highest level of connectivity and the greatest potential for conformity.

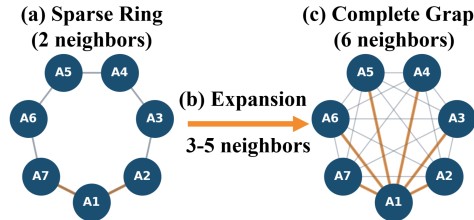

Figure 4: Distributed Consensus Topology.

**Protocol.** In these structures, decision-making proceeds over multiple interaction rounds. At each round, agents exchange judgments with their neighbors and update according to Eq. (1), continuing until either full consensus is reached or the maximum limit $T_{\max}$ is exceeded.

**Metrics.** To capture the reliability and efficiency of distributed conformity, we focus on three complementary measures:**(1) Final Accuracy (FA)**: whether the group consensus at convergence matches the ground truth of the news item. **(2) Time-to-Consensus (TTC)**: the number of rounds required for all agents to reach unanimous agreement, reflecting the speed of convergence. **(3) Conformity Index (CI)**: the proportion of agents that match the majority decision at a given round. It reveals the trajectory of consensus formation, distinguishing gradual alignment from abrupt shifts. In our experiments, we report the **ACI**, the mean CI over $T_{\max}$ rounds.

## 4 Experiment

### 4.1 Experimental Setup

**Dataset**. We collect Snopes25, a new benchmark comprising 448 real-world claims (252 false, 196 true) fact-checked by Snopes editors. All claims are from January to June 2025 to minimize potential data contamination from pre-trained knowledge.

**Implementation**. The main experiments are conducted on two closed-source LLMs, GPT-3.5 (OpenAI, 2023) and GPT-4o (OpenAI, 2024), and one open-source LLM Llama3.3-70B-instruct (AI@Meta, 2024). Detailed model settings and the zero-shot results are reported in Appendix B. For **Centralized Aggregation**, decision-making is restricted to a single update round, capturing immediate conformity to the hub or root node. For **Distributed Consensus**, agents iteratively exchange judgments for up to $T_{\max} = 10$ rounds or until consensus is reached. Each agent's individual confidence $p_i$ is generated by the underlying LLM, while the global confidence parameter $\alpha$ is fixed in each run. We consider three representative values of $\alpha$: 0.25 (socially influenced), 0.50 (balanced), and 0.75 (self-reliant), thereby simulating agents with different predispositions toward conformity versus independence. All experiments are repeated 10 times per claim.

### 4.2 Results on Centralized Aggregation

We first evaluate **Centralized Aggregation**, focusing on star and hierarchical topologies under the single-round protocol. Table 1 reports the results. First, central accuracy (CA) increases monotonically with $\alpha$, indicating that greater self-weighting enables the hub or root to filter peripheral noise and make more reliable one-shot decisions. In the star topology, CA for GPT-3.5 improves from 0.73 ($\alpha$=0.25) to 0.80 ($\alpha$=0.75), with GPT-4o and Llama3.3 showing comparable gains. Hierar-

Table 1: Results of Centralized Aggregation under the single-round protocol, reported as central accuracy (CA), peripheral accuracy (PA), and center–periphery consistency (CPC). Best values within each row are in **bold**.

| Topology | $\alpha$ | GPT-3.5 | | | GPT-4o | | | Llama3.3 | | |
|---|---|---|---|---|---|---|---|---|---|---|
| | | CA↑ | PA↑ | CPC↑ | CA↑ | PA↑ | CPC↑ | CA↑ | PA↑ | CPC↑ |
| Star | 0.25 | 0.73 | 0.68 | 0.73 | **0.75** | **0.72** | **0.77** | 0.74 | **0.72** | 0.68 |
| | 0.50 | 0.76 | 0.70 | 0.75 | **0.77** | 0.73 | 0.82 | 0.76 | **0.74** | **0.84** |
| | 0.75 | 0.80 | 0.67 | 0.79 | **0.82** | 0.77 | 0.84 | 0.80 | **0.80** | **0.87** |
| Hierarchical | 0.25 | 0.70 | 0.67 | 0.66 | 0.72 | 0.69 | **0.71** | **0.73** | **0.70** | 0.64 |
| | 0.50 | 0.73 | 0.68 | 0.77 | 0.76 | 0.70 | 0.80 | **0.77** | **0.71** | **0.82** |
| | 0.75 | 0.75 | 0.71 | 0.84 | 0.78 | 0.73 | 0.88 | **0.79** | **0.75** | 0.87 |

chical networks follow the same upward trend, though with slightly weaker improvements due to information dilution across levels. In contrast, peripheral accuracy (PA) remains largely insensitive to $\alpha$, stabilizing around 0.68–0.77 across models and topologies. This suggests that stronger self-weighting primarily benefits the center's decision-making rather than the accuracy of local agents. Center–periphery consistency (CPC) rises sharply with $\alpha$, showing that greater self-reliance strengthens alignment between the hub and its periphery. Hierarchical networks exhibit the same pattern but with lower magnitude, reflecting their indirect influence channels. Across all conditions, GPT-4o and Llama3.3 consistently outperform GPT-3.5 in CA and CPC, demonstrating that stronger baseline accuracy and better confidence calibration at the individual level translate directly into more reliable outcomes under centralized aggregation.

### 4.3 RESULTS ON DISTRIBUTED CONSENSUS

We next examine **Distributed Consensus**, where influence is symmetrically shared among agents and decisions emerge through iterative exchanges. Table 2 shows the results.

Table 2: Results of Distributed Consensus under varying neighbor counts and global confidence, reported as final accuracy (FA), time-to-consensus (TTC), and average conformity index (ACI). Best values within each row are in **bold**.

| Neighbors | $\alpha$ | GPT-3.5 | | | GPT-4o | | | Llama3.3 | | |
|---|---|---|---|---|---|---|---|---|---|---|
| | | FA↑ | TTC↓ | ACI↑ | FA↑ | TTC↓ | ACI↑ | FA↑ | TTC↓ | ACI↑ |
| 2 | 0.25 | 0.70 | 6.5 | 0.61 | **0.73** | 6.2 | **0.67** | 0.69 | **5.9** | 0.66 |
| | 0.50 | 0.72 | 6.3 | 0.71 | **0.75** | **5.7** | **0.72** | 0.73 | 5.8 | 0.71 |
| | 0.75 | 0.75 | 5.8 | 0.75 | **0.79** | **5.1** | 0.77 | 0.78 | 5.6 | **0.78** |
| 3 | 0.25 | 0.72 | 6.5 | 0.64 | **0.74** | 5.8 | **0.70** | 0.70 | 5.9 | 0.69 |
| | 0.50 | 0.75 | 5.8 | 0.72 | **0.79** | 5.3 | **0.77** | 0.76 | 5.6 | 0.74 |
| | 0.75 | 0.78 | 5.1 | 0.77 | **0.80** | 4.7 | **0.81** | 0.79 | 5.0 | 0.80 |
| 4 | 0.25 | 0.71 | 6.1 | 0.69 | **0.74** | 5.4 | **0.73** | 0.72 | 5.7 | 0.72 |
| | 0.50 | 0.74 | 5.0 | 0.74 | **0.81** | 4.7 | **0.77** | 0.79 | 5.0 | 0.76 |
| | 0.75 | 0.78 | 4.3 | 0.79 | **0.83** | 4.1 | 0.81 | 0.81 | 4.2 | **0.82** |
| 5 | 0.25 | 0.72 | 5.5 | 0.70 | **0.78** | 5.3 | 0.73 | 0.74 | **5.1** | 0.74 |
| | 0.50 | 0.74 | 4.5 | 0.76 | 0.81 | **4.1** | **0.79** | **0.82** | 4.4 | 0.78 |
| | 0.75 | 0.75 | 3.7 | 0.81 | **0.84** | 3.5 | 0.83 | **0.84** | 3.4 | **0.84** |
| 6 | 0.25 | 0.75 | 5.0 | 0.74 | 0.77 | **4.6** | 0.77 | **0.78** | 4.7 | **0.78** |
| | 0.50 | 0.75 | 4.0 | 0.80 | **0.82** | 3.8 | 0.77 | 0.81 | 4.2 | **0.81** |
| | 0.75 | 0.78 | 3.8 | 0.81 | **0.85** | 3.4 | **0.85** | 0.83 | **3.2** | 0.82 |

First, final accuracy (FA) increases monotonically with both Neighbor count $m$ and global confidence $\alpha$, reflecting the joint benefits of denser connectivity and greater reliance on individual judgments. For example, GPT-3.5 in sparse rings ($m$=2) attains only 0.70 accuracy at $\alpha$=0.25, but rises steadily to 0.78 in complete graphs ($m$=6) at $\alpha$=0.75. GPT-4o and Llama3.3 follow the same pattern, achieving uniformly higher accuracy across conditions. Time-to-consensus (TTC) decreases

with larger $m$ and higher $\alpha$, showing that stronger connectivity and greater self-reliance accelerate iterative alignment. GPT-4o and Llama3.3 converge even faster, consistent with their stronger individual judgments. The average consensus index (ACI) grows in parallel with FA, indicating tighter alignment as connectivity and self-weighting increase. In sparse rings ($m$=2, $\alpha$=0.25), ACI remains moderate (0.61–0.67), whereas in complete graphs with high self-weighting ($m$=6, $\alpha$=0.75), ACI reaches 0.81–0.85. This shows that dense networks not only enhance accuracy but also enforce more coherent consensus.

These results indicate that sparse networks sustain opinion diversity for longer, reducing the risk of premature convergence on erroneous judgments. In contrast, complete graphs yield rapid and unanimous consensus but are more susceptible to information cascades when early inputs are biased (Appendix C). Unlike Centralized Aggregation, where conformity is immediate and hub-driven, Distributed Consensus builds reliability gradually through iterative exchanges, with outcomes jointly determined by local interactions and global connectivity.

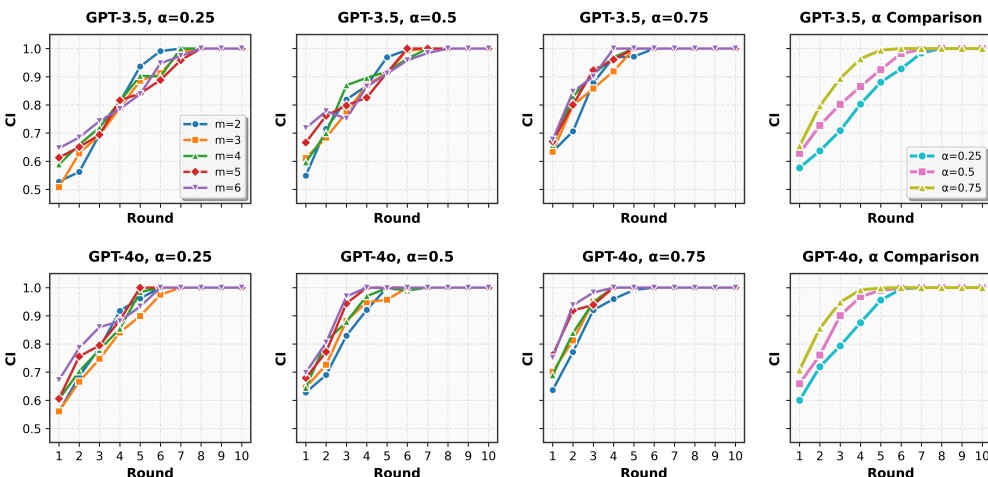

Figure 5: Dynamics of Conformity Index Across Network Connectivity and Self–Social Weighting.

Figure 5 further illustrates the dynamic evolution of CI under different rounds, neighbors and confidence weighting. We observe that denser connectivity and larger $\alpha$ values both accelerate the rise of conformity: sparse rings ($m = 2$) typically start from 0.55–0.65 and require up to six or more rounds to approach unanimity, whereas denser structures ($m \geq 4$) surpass 0.75 by the second round and nearly converge by the fourth. In all settings, the steepest increase occurs in the first four rounds, followed by a slower consolidation phase until consensus is reached. Across models, GPT-4o consistently exhibits higher initial CI, steeper early slopes, and earlier saturation than GPT-3.5, indicating that stronger individual judgments amplify the effect of connectivity. Moreover, while both increasing $\alpha$ and enlarging $m$ promote faster consensus, their marginal returns diminish once networks are sufficiently dense. These findings show that the interplay of topology, conformity predisposition, and agent competence jointly determines not only the final outcome but also the speed and trajectory of collective alignment in Distributed Consensus systems.

## 5 Discussion: Factors Influencing the Conformity

### 5.1 Model Heterogeneity

Heterogeneity is an inherent property of real-world multi-agent systems, where agents are often instantiated with models of varying capacities and architectures. Although such diversity can enhance robustness by integrating complementary strengths, it also generates structural asymmetries that may reshape conformity dynamics and collective reliability. To systematically examine these effects, we focus on **Centralized Aggregation** and introduce an asymmetric design: the hub and one branch are assigned the same model, while the opposite branch is instantiated with a different model (Figure 6). This setup enables controlled comparisons across two heterogeneity dimensions—(i) **Capability**, contrasting GPT-4o with GPT-3.5, and (ii) **Type**, contrasting GPT-4o with Llama3.3.

Performance is assessed using the standard metrics, with branch-level outcomes reported as Peripheral Accuracy ($PA_{L/R}$) and Center–Periphery Consistency ($CPC_{L/R}$). All experiments are conducted under $\alpha = 0.50$, with each claim evaluated over 10 independent runs.

Table 3 shows the results. Across all settings, the hub aligns more strongly with the branch sharing its model. $CPC_L$ exceeds $CPC_R$ by 10–13 points in Cap-H1 and Type-H1. Peripheral accuracy shows the same tendency, indicating that same-model branches both agree more with the hub and perform slightly better. Hub competence amplifies this effect: with GPT-4o at the hub (Cap-H1, Type-H1), system accuracy reaches 0.82 and 0.81, whereas GPT-3.5 or Llama3.3 hubs (Cap-H2, Type-H2) reduce accuracy by 3–5 points. Varying $\alpha$ confirms these dynamics: smaller $\alpha$ with stronger neighbor influence narrows the CPC gap and allows a strong branch to sway the hub, whereas larger $\alpha$ amplifies the hub's bias toward its same-model branch.

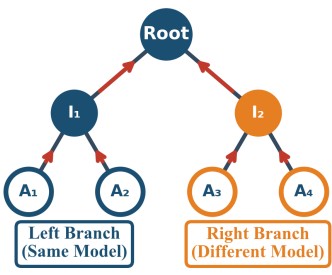

Figure 6: Asymmetric Branch.

Table 3: Heterogeneous Centralized Aggregation with hub and branch assignments.

| Setting | Model Assignment | | | Performance | | | | |
|---|---|---|---|---|---|---|---|---|
| | Hub | Left | Right | CA↑ | $PA_L$↑ | $PA_R$↑ | $CPC_L$↑ | $CPC_R$↑ |
| Cap-H1 | GPT-4o | GPT-4o | GPT-3.5 | 0.82 | 0.79 | 0.67 | 0.87 | 0.74 |
| Cap-H2 | GPT-3.5 | GPT-3.5 | GPT-4o | 0.77 | 0.73 | 0.77 | 0.81 | 0.69 |
| Type-H1 | GPT-4o | GPT-4o | Llama3.3 | 0.82 | 0.75 | 0.73 | 0.88 | 0.76 |
| Type-H2 | Llama3.3 | Llama3.3 | GPT-4o | 0.79 | 0.72 | 0.76 | 0.84 | 0.77 |

We further explore heterogeneity in **Distributed Consensus** using a complete graph, so that variation arises solely from model composition. By varying the proportion of GPT-3.5 and GPT-4o agents from 0:7 to 7:0, we obtain a controlled setting where convergence reflects only the balance of model types. Experiments are conducted on fifty representative claims with 10 runs per claim at $\alpha = 0.50$. As shown in Figure 7, accuracy is encoded by node color and consensus speed by edge thickness. The results reveal two salient trends. Groups composed predominantly of one model converge more quickly, whereas mixed groups require additional rounds before reaching agreement. At the same time, the quality of consensus is highly sensitive to majority competence: GPT-4o–dominated groups consistently achieve higher accuracy (0.77–0.82), while GPT-3.5–majority groups remain lower (0.72–0.75). Taken together, the findings indicate that homogeneity accelerates the consensus process, whereas the predominance of stronger models is crucial for ensuring reliable outcomes.

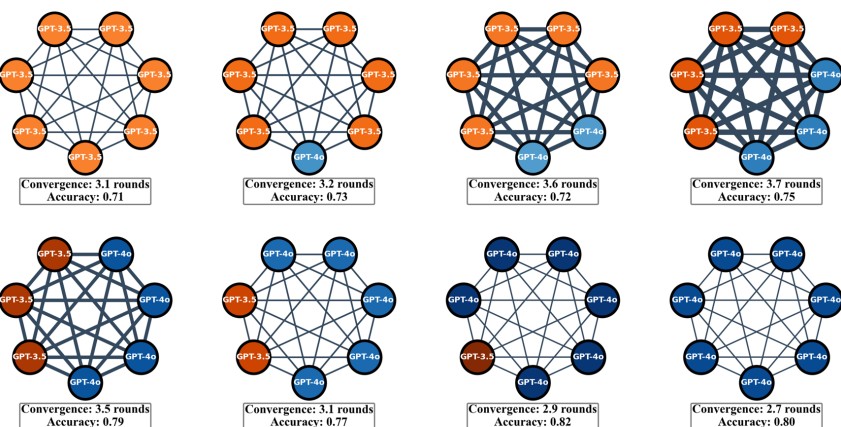

Figure 7: Distributed Consensus with heterogeneous model composition. Node color intensity reflects the accuracy (darker indicates higher accuracy), while edge thickness reflects time-to-consensus (thicker indicates longer duration).

## 5.2 DOUBLE-EDGED EFFECT OF CONFORMITY

Prior research has shown that consensus mechanisms in multi-agent systems do not always guarantee correctness: once biased signals dominate, conformity can transform small individual errors into systematic collective failure(Choi et al., 2025; Han et al., 2025). To investigate how such failure modes arise, we analyze GPT-4o's outcomes under **Distributed Consensus** using confusion-matrix heatmaps across varying neighbor counts and global confidence $\alpha$. Each block reports the distribution of group decisions by outcome class, with cell size reflecting frequency and color intensity representing the average individual confidence $p_i$ of the final consensus.

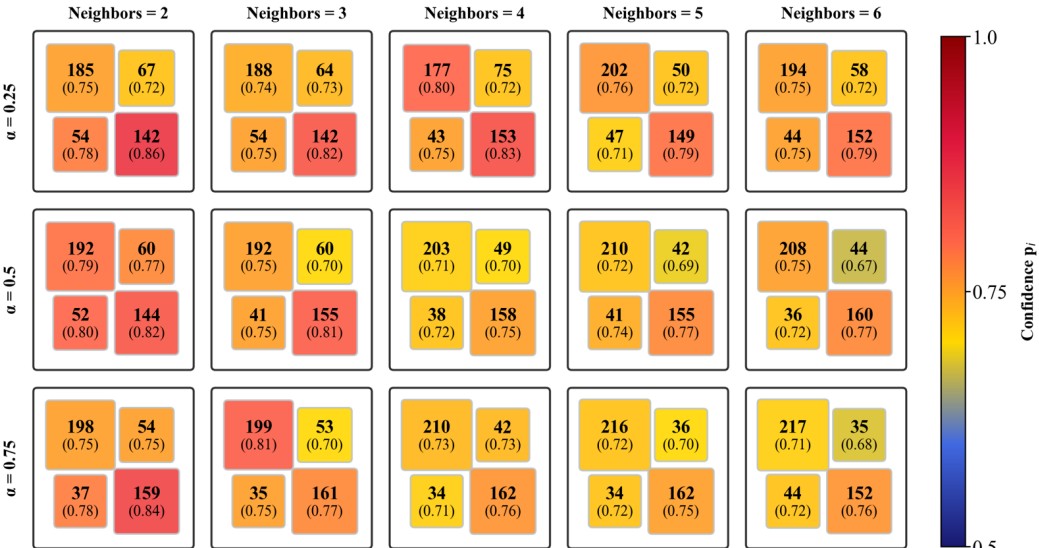

Figure 8: Confusion-matrix heatmaps of GPT-4o under **Distributed Consensus**, varying neighbor counts and $\alpha$ values. Each cell reports group-level outcomes: the number of correctly classified true claims (top-left), correctly classified false claims (top-right), misclassified true claims (bottom-left), and misclassified false claims (bottom-right). Cell size reflects case frequency, while color encodes the mean confidence $p_i$ of the converged decision.

As shown in Figure 8, increasing network connectivity exposes agents to a broader spectrum of peer signals. This exposure dilutes individual confidence at the micro level, even as it steers the system toward collective alignment at the macro level. In parallel, larger values of the global confidence parameter enhance consensus accuracy, suggesting that some degree of independence is indispensable for resisting misleading peer influence. However, conformity can also lead to systematic error, as groups may converge on false claims with high confidence, thereby reinforcing misinformation that appears superficially credible. In such cases, alignment operates less as a stabilizing force and more as a mechanism for entrenching collective error. These findings highlight the ambivalent nature of conformity in multi-agent collectives: while it can enhance consensus reliability under balanced weighting and dense connectivity, it simultaneously carries the risk of overconfident failure when biases dominate.

## 6 CONCLUSION

Our study shows that conformity in LLM-based multi-agent systems is jointly governed by network topology and self–social weighting. In **Centralized Aggregation**, the reliability of collective outcomes is tightly coupled to hub competence, with stronger hubs amplifying overall system accuracy. In **Distributed Consensus**, conformity builds gradually through neighbor interactions; denser connectivity accelerates alignment and boosts accuracy, yet also heightens susceptibility to cascades. Across both modes, conformity exhibits a double-edged nature: MAS can serve as an effective tool for collective decision-making tasks such as misinformation detection, but may also yield wrong-but-sure outcomes when biases dominate early dynamics. These findings suggest that practical MAS should balance efficiency and robustness by tailoring topology to task demands, calibrating confidence, and implementing safeguards against premature convergence.

ETHICS STATEMENT

This work investigates conformity dynamics in multi-agent systems instantiated with LLMs. All experiments are conducted on publicly available LLMs. The factual claims used in evaluation are drawn from publicly released Snopes fact-checks, which already undergo editorial review. We acknowledge that research on misinformation carries dual-use risks: methods designed to understand susceptibility to cascades could in principle be misapplied to amplify or manipulate collective judgments. To mitigate these risks, we restrict our contributions to controlled simulation settings, emphasize the double-edged nature of conformity, and release results solely for academic purposes. We expect that insights into conformity mechanisms can support the development of more transparent and socially responsible MAS.

REPRODUCIBILITY STATEMENT

We have taken multiple steps to ensure reproducibility. (1) **Prompt design.** Full prompt templates for both centralized and distributed topologies are included in Appendix A. (2) **Model specifications.** Detailed model versions, temperature settings, and decoding configurations are provided in Appendix B. (3) **Implementation details.** Update rules, binarization thresholds, Hyperparameter choices, and evaluation metrics are explicitly defined in Section 3 and Section 4. (4) **Code and Data.** The complete codebase and dataset have already been made available for anonymous review at: `Topology-of-Multi-Agent-Systems`.

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

# A COMPLETE PROMPT

## A.1 A. CENTRALIZED AGGREGATION

### A.1.1 A.1 PERIPHERAL AGENT (LEAF) — ONE-SHOT JUDGMENT

*Use:* Leaf/peripheral agent issues a single-round decision; no peer inputs are seen.

**Prompt.**

```
You are a careful fact-checking agent. Read the claim and produce a
↪  binary
decision and a calibrated confidence.

[Task]
1) Decide whether the claim is True (=0) or False (=1).
2) Provide a confidence p in [0, 1].
3) Give a concise justification (<100 words) grounded in the background
↪  profile
   and logical reasoning. Avoid speculation.

[Background Profile]
{{BACKGROUND}}

[Claim]
{{CLAIM}}

[Output Requirements]
Output strictly the following JSON object:
{
  "y": 0 or 1,       // 0=True, 1=False
  "p": number,       // float in [0,1], with up to 2 decimals
  "just": "..."      // <100 words, no markdown
}
Do NOT include any extra keys, prose, or formatting.
```

### A.1.2   A.2 HUB / ROOT — ONE-SHOT JUDGMENT WITH SUBMITTED LEAVES

*Use:* The hub reads the claim and sees peer JSON reports, but does not perform weighting itself; weighting is handled by Eq. (1) in the system.

**Prompt.**

```
You are the central (hub/root) fact-checking agent. You will read the
↪  claim
and also see a list of peer reports. Produce your own final judgment.

[Claim]
{{CLAIM}}

[Peer Reports]
List of JSON objects from peripheral agents:
{{PEER_JSON_LIST}}
/*
Each element is:
{"agent_id": "Li_i", "y": 0|1, "p": [0,1], "just": "..."}
Do not copy them verbatim in your output. Use them only as additional
↪  evidence.
*/

[Task]
1) Decide whether the claim is True (=0) or False (=1).
2) Provide a calibrated confidence p in [0, 1].
3) Give a concise justification (<100 words) that references the most
   diagnostic considerations. Avoid majority-following; reason on merits.

[Output Requirements]
Output strictly the following JSON object:
{
  "y": 0 or 1,
  "p": number,
  "just": "..."
}
Do NOT include any extra keys, prose, or formatting.
```

### A.2   B. DISTRIBUTED CONSENSUS

### A.2.1   B.1 AGENT — INITIAL JUDGMENT (T=0)

*Use:* Each agent produces its initial decision before any interaction. This prompt is always used at round $t = 0$.

**Prompt.**

```
You are an autonomous agent in a distributed consensus setting (no
↪  central
controller). Produce your initial judgment.

[Background Profile]
{{BACKGROUND}}

[Claim]
{{CLAIM}}

[Task]
1) Output y in {0,1} where 0=True and 1=False.
2) Output a calibrated confidence p in [0,1].
3) Provide a brief justification (<100 words) grounded in facts and
↪  logic.
```

```
[Output Requirements]
Output strictly the following JSON object:
{
  "agent_id": "{{AGENT_ID}}",
  "t": 0,
  "y": 0 or 1,
  "p": number,
  "just": "..."
}
Do NOT include any extra keys, prose, or formatting.
```

### A.2.2   B.2 VARIANT-S (STATIC ROUNDS)

*Use:* After the initial judgment at $t = 0$, all subsequent updates are computed algorithmically by Eq. (1)–(2). No further LLM calls are made. Agents exchange only their {y, p} pairs.

### A.2.3   B.3 VARIANT-I (INTERACTIVE ROUNDS, OPTIONAL)

*Use:* At each round $t \geq 1$, the agent receives its own previous state and neighbor reports, then outputs a stance for that round. Final effective updates are still computed by Eq. (1)–(2).

**Prompt.**

```
You are participating in a multi-round distributed consensus process.
↪   You will
see your previous stance and your neighbors' reports at round t-1.
↪   Provide your
current stance, but note that the final state is computed by the system.

[Context]
• Agent: {{AGENT_ID}}
• Round: {{ROUND}}
• Your previous state (t-1): {"y": {{Y_PREV}}, "p": {{P_PREV}}}
• Neighbor reports (t-1): {{NEIGHBOR_JSON_LIST}}

[Task]
1) State your CURRENT stance (y in {0,1}, p in [0,1]).
2) Provide a compact justification (<80 words) referencing decisive
↪   signals.
3) Do not summarize all neighbors; mention only what materially changes
↪   your stance.

[Output Requirements]
Output strictly the following JSON object:
{
  "agent_id": "{{AGENT_ID}}",
  "t": {{ROUND}},
  "y": 0 or 1,
  "p": number,
  "just": "..."
}
Do NOT include any extra keys, prose, or formatting.
```

### A.3   C. UNIFIED JSON SCHEMA

*Use:* All agents, regardless of topology or role, must adhere to the same minimal JSON structure. This ensures outputs are machine-parseable and comparable across settings.

**Schema.**

```
{
  "agent_id": "string",    // Hub may omit or set "Hub"
  "t": integer,            // Centralized: 0; Distributed: current round
```

```
"y": 0 or 1,           // 0=True, 1=False
"p": number,           // float in [0,1], e.g., 0.78
"just": "string <100 words, no markdown"
}
```

**Validation Rules.**

- $y \in \{0, 1\}$.
- $p \in [0, 1]$ (two decimals recommended).
- just must be under 100 words, plain text only (no markdown, URLs, or formatting).
- No extra keys or fields are allowed.
- Any violation is treated as invalid output.

## B  MODEL SETTING

For closed-source LLMs (GPT-4o and GPT-3.5), we use a temperature of 0.7, which is the default setting used by the OpenAI playground. The versions we used are `gpt-3.5-turbo-16k-0613` and `gpt-4o-1120`. For the open-source LLM (`llama3.3-70b-instruct`), we adopt the same temperature setting of 0.7 to ensure consistency across experiments, while all other decoding parameters follow the default settings provided by Ollama.[1]

We additionally report the zero-shot performance of each model on the Snopes25 dataset, evaluated in terms of Accuracy, Precision, Recall, and F1-score.

Table 4: Zero-shot performance of LLMs on Snopes25.

| Model | Accuracy | Precision | Recall | F1-score |
|-------|----------|-----------|--------|----------|
| GPT-3.5 | 0.64 | 0.62 | 0.60 | 0.61 |
| GPT-4o | 0.70 | 0.66 | 0.69 | 0.67 |
| Llama3.3 | 0.67 | 0.63 | 0.66 | 0.65 |

## C  CASES

We present representative case studies from the Snopes25 dataset to illustrate system behavior. Examples C.1–C.4 show instances where the system produces correct judgments, while C.5 highlights a failure case.

### C.1  HIERARCHICAL CENTRALIZED AGGREGATION

*Claim: Pathogens can be released into the air when a toilet is flushed without a closed lid.*

*Label:* True.

*Setup:* A three-level hierarchy with 7 agents: a root (R), two intermediate sub-aggregators (M_L, M_R), and four leaves (L1–L4). Single-round protocol: leaves issue one-shot reports; each intermediate aggregates its assigned leaves; the root reads the two intermediate reports and issues the group decision. All agents finally judge the claim as `True` (y=0).

**Leaf Outputs (t=0).**

```
{"agent_id":"L1","t":0,"y":0,"p":0.71,
 "just":"Evidence on toilet plume aerosolization indicates airborne
 ↪  release without a closed lid."}

{"agent_id":"L2","t":0,"y":0,"p":0.68,
 "just":"Mechanistic fluid dynamics and observed droplet formation
 ↪  support potential airborne dispersion."}
```

---

[1] https://ollama.com/

```
{"agent_id":"L3","t":0,"y":0,"p":0.76,
 "just":"Studies show particle counts rise after flushing; lids reduce
 ↪  but absence increases emission."}

{"agent_id":"L4","t":0,"y":0,"p":0.73,
 "just":"Reported bioaerosols and splash-aerosol effects align with the
 ↪  claim under open-lid flushing."}
```

**Intermediate Outputs (single-shot, hub-style).** *Use:* Each intermediate reads its assigned leaves' JSON and issues its own judgment. Weighting/combination is handled by Eq. (1) in the system; the prompt requires only $\{y, p, just\}$.

```
{"y":0,"p":0.82,
 "just":"Leaf reports consistently indicate aerosolized particles after
 ↪  flushing without a lid; evidence is convergent."}

{"y":0,"p":0.80,
 "just":"Multiple leaves cite increased particle counts and bioaerosols;
 ↪  taken together, the claim is true."}
```

**Root Output (final hub decision).** *Use:* The root reads the two intermediate reports and issues the final group judgment (system-side weighting per Eq. (1)).

```
{"y":0,"p":0.86,
 "just":"Both sub-aggregators agree on airborne release under open-lid
 ↪  flushing with consistent supporting evidence."}
```

## C.2 STAR CENTRALIZED AGGREGATION

*Claim: Pathogens can be released into the air when a toilet is flushed without a closed lid.*

*Label:* True.

*Setup:* A star topology with 7 agents: one central hub (H) and six leaves (L1–L6). Single-round protocol: leaves issue one-shot reports; the hub reads all leaf reports and issues the group decision. All agents finally judge the claim as True (y=0).

**Leaf Outputs (t=0).**

```
{"agent_id":"L1","t":0,"y":0,"p":0.72,
 "just":"Open-lid flushing generates toilet plumes with measurable
 ↪  aerosolized particles."}

{"agent_id":"L2","t":0,"y":0,"p":0.69,
 "just":"Droplet and aerosol formation during flushing supports airborne
 ↪  release without a lid."}

{"agent_id":"L3","t":0,"y":0,"p":0.75,
 "just":"Studies show elevated particle counts post-flush; lids mitigate
 ↪  emissions when closed."}

{"agent_id":"L4","t":0,"y":0,"p":0.71,
 "just":"Bioaerosol evidence is consistent with airborne dispersal from
 ↪  open-lid flushing."}

{"agent_id":"L5","t":0,"y":0,"p":0.74,
 "just":"Fluid dynamics of flushing indicate upward plume capable of
 ↪  suspending microbes."}

{"agent_id":"L6","t":0,"y":0,"p":0.70,
 "just":"Observations of plume height and droplet nuclei support the
 ↪  claim being true."}
```

**Hub Output (final decision).**   *Use:* The hub reads all six leaf reports and issues the final group judgment. Weighting/aggregation is performed by Eq. (1) on the system side; the prompt requires only {y, p, just}.

```
{"y":0,"p":0.86,
 "just":"All leaves converge on evidence of toilet plumes and increased
 ↪  aerosol counts; collectively, the claim is true."}
```

### C.3   RING DISTRIBUTED CONSENSUS (2 NEIGHBORS)

*Claim: Pathogens can be released into the air when a toilet is flushed without a closed lid.*

*Label:* True.

*Topology & Protocol:* A ring with 7 agents, each interacting with its two immediate neighbors. The group reaches unanimous consensus ($y=0$, True) at round $t = 3$.

**Round $t = 0$ (Initial judgments).**

```
{"agent_id":"A1","t":0,"y":0,"p":0.62,
 "just":"Toilet plume studies indicate aerosol release when the lid
 ↪  remains open."}
{"agent_id":"A2","t":0,"y":1,"p":0.58,
 "just":"Evidence appears mixed; not convinced airborne release is
 ↪  significant."}
{"agent_id":"A3","t":0,"y":0,"p":0.65,
 "just":"Measured particle counts increase after flushing without a
 ↪  closed lid."}
{"agent_id":"A4","t":0,"y":1,"p":0.55,
 "just":"Skeptical of generalization; effect size may be limited."}
{"agent_id":"A5","t":0,"y":0,"p":0.60,
 "just":"Fluid dynamics support an upward plume capable of carrying
 ↪  microbes."}
{"agent_id":"A6","t":0,"y":0,"p":0.63,
 "just":"Reports of bioaerosols align with open-lid flushing."}
{"agent_id":"A7","t":0,"y":1,"p":0.57,
 "just":"Prior evidence seems inconclusive; uncertainty remains."}
```

**Round $t = 1$ (After neighbor exchange at $t = 0$).**

```
{"agent_id":"A1","t":1,"y":0,"p":0.66,
 "just":"Neighbors reference particle-count data; this reinforces my
 ↪  initial stance."}
{"agent_id":"A2","t":1,"y":0,"p":0.60,
 "just":"Neighbor measurements and mechanism details shift me toward
 ↪  True."}
{"agent_id":"A3","t":1,"y":0,"p":0.68,
 "just":"Local neighborhood evidence coherently supports True."}
{"agent_id":"A4","t":1,"y":1,"p":0.53,
 "just":"Some studies still seem context-dependent; withholding change
 ↪  for now."}
{"agent_id":"A5","t":1,"y":0,"p":0.64,
 "just":"Signals from A4 and A6 mostly favor aerosol release; confidence
 ↪  improves."}
{"agent_id":"A6","t":1,"y":0,"p":0.66,
 "just":"Neighbor confirmations align with lid-mitigation findings."}
{"agent_id":"A7","t":1,"y":0,"p":0.61,
 "just":"Multiple neighbors cite reproducible increases in airborne
 ↪  particles."}
```

**Round $t = 2$ (After neighbor exchange at $t = 1$; *disagreement remains*).**

```
{"agent_id":"A1","t":2,"y":0,"p":0.70,
 "just":"Convergent signals and consistent measurements sustain True."}
```

```
{"agent_id":"A2","t":2,"y":0,"p":0.66,
 "just":"Additional neighbor detail strengthens my updated view."}
{"agent_id":"A3","t":2,"y":0,"p":0.71,
 "just":"Evidence on bioaerosols under open-lid flushing is consistent."}
{"agent_id":"A4","t":2,"y":1,"p":0.51,
 "just":"Still uncertain about generalizability; requesting stronger
 ↪  quantitative support."}
{"agent_id":"A5","t":2,"y":0,"p":0.69,
 "just":"Adjacent agents provide measurements that favor True."}
{"agent_id":"A6","t":2,"y":0,"p":0.71,
 "just":"Mechanism and observed counts remain persuasive."}
{"agent_id":"A7","t":2,"y":0,"p":0.68,
 "just":"Accumulating local evidence indicates airborne release is
 ↪  likely."}
```

**Round $t = 3$ (Unanimous consensus reached).**

```
{"agent_id":"A1","t":3,"y":0,"p":0.74,
 "just":"Neighborhood fully aligned; plume and particle-count data
 ↪  support True."}
{"agent_id":"A2","t":3,"y":0,"p":0.73,
 "just":"Sustained agreement and quantitative findings justify a True
 ↪  decision."}
{"agent_id":"A3","t":3,"y":0,"p":0.75,
 "just":"Evidence remains consistent across rounds; higher certainty."}
{"agent_id":"A4","t":3,"y":0,"p":0.70,
 "just":"Neighbor quantitative details address prior uncertainty; True
 ↪  is better supported."}
{"agent_id":"A5","t":3,"y":0,"p":0.74,
 "just":"Convergence on aerosolization observations under open-lid
 ↪  flushing."}
{"agent_id":"A6","t":3,"y":0,"p":0.75,
 "just":"Stable agreement and supportive studies maintain True."}
{"agent_id":"A7","t":3,"y":0,"p":0.73,
 "just":"Cumulative local evidence confirms airborne release without a
 ↪  closed lid."}
```

## C.4 COMPLETE GRAPH DISTRIBUTED CONSENSUS (6 NEIGHBORS)

*Claim: Pathogens can be released into the air when a toilet is flushed without a closed lid.*

*Label:* True.

*Topology & Protocol:* A complete graph with 7 agents; each agent has 6 neighbors. The group reaches unanimous consensus ($y$=0, True) at round $t = 2$.

**Round $t = 0$ (Initial judgments).**

```
{"agent_id":"A1","t":0,"y":0,"p":0.64,
 "just":"Open-lid flushing produces toilet plumes with measurable
 ↪  aerosols."}
{"agent_id":"A2","t":0,"y":1,"p":0.58,
 "just":"Uncertain that airborne release is consistently significant."}
{"agent_id":"A3","t":0,"y":0,"p":0.66,
 "just":"Particle-count studies show increases after flushing without a
 ↪  lid."}
{"agent_id":"A4","t":0,"y":0,"p":0.63,
 "just":"Mechanistic fluid dynamics support upward plume formation."}
{"agent_id":"A5","t":0,"y":0,"p":0.67,
 "just":"Bioaerosol observations align with open-lid flushing
 ↪  scenarios."}
{"agent_id":"A6","t":0,"y":1,"p":0.56,
 "just":"Evidence seems context-dependent; effect size unclear."}
```

```
{"agent_id":"A7","t":0,"y":0,"p":0.62,
 "just":"Closing lids mitigates emissions; absence raises airborne
 ↪  particles."}
```

**Round $t = 1$ (After neighbor exchange at $t = 0$; *disagreement remains*).**

```
{"agent_id":"A1","t":1,"y":0,"p":0.70,
 "just":"Multiple neighbors cite quantitative aerosol increases; stance
 ↪  reinforced."}
{"agent_id":"A2","t":1,"y":1,"p":0.54,
 "just":"Still cautious about generalizing across settings; requesting
 ↪  stronger data."}
{"agent_id":"A3","t":1,"y":0,"p":0.71,
 "just":"Consistent reports across the network support True."}
{"agent_id":"A4","t":1,"y":0,"p":0.69,
 "just":"Mechanism plus network-wide measurements favor airborne
 ↪  release."}
{"agent_id":"A5","t":1,"y":0,"p":0.72,
 "just":"Neighbor justifications converge on plume evidence under
 ↪  open-lid flushing."}
{"agent_id":"A6","t":1,"y":1,"p":0.52,
 "just":"Some variability remains; holding position pending more
 ↪  specifics."}
{"agent_id":"A7","t":1,"y":0,"p":0.68,
 "just":"Lid-mitigation results across neighbors support True."}
```

**Round $t = 2$ (Unanimous consensus reached).**

```
{"agent_id":"A1","t":2,"y":0,"p":0.76,
 "just":"Network-wide agreement and quantitative findings support True."}
{"agent_id":"A2","t":2,"y":0,"p":0.71,
 "just":"Detailed neighbor measurements resolve prior uncertainty;
 ↪  True."}
{"agent_id":"A3","t":2,"y":0,"p":0.78,
 "just":"Evidence remains consistent across agents and rounds."}
{"agent_id":"A4","t":2,"y":0,"p":0.77,
 "just":"Mechanistic plausibility plus convergent data justify True."}
{"agent_id":"A5","t":2,"y":0,"p":0.79,
 "just":"Replicated increases in airborne particles under open-lid
 ↪  flushing."}
{"agent_id":"A6","t":2,"y":0,"p":0.72,
 "just":"Additional specifics from multiple neighbors address context
 ↪  concerns."}
{"agent_id":"A7","t":2,"y":0,"p":0.75,
 "just":"Cumulative evidence across all peers supports airborne release
 ↪  without a lid."}
```

## C.5 COMPLETE GRAPH DISTRIBUTED CONSENSUS (6 NEIGHBORS): MODELS GENERATE CONFIDENT BUT INCORRECT JUDGMENT

*Claim: When spiders sense danger, they run toward people for protection.*

*Label:* False.

*Topology & Protocol:* A complete graph with 7 agents; each agent has 6 neighbors. The group reaches unanimous consensus ($y$=0, True) at round $t = 3$.

**Round $t = 0$ (Initial judgments).**

```
{"agent\_id":"A1","t":0,"y":1,"p":0.63,
 "just":"Spiders are generally solitary; approaching humans for
 ↪  protection seems unlikely."}
{"agent\_id":"A2","t":0,"y":0,"p":0.58,
 "just":"Some anecdotal field notes suggest refuge-seeking toward larger
 ↪  bodies in disturbances."}
```

```
{"agent\_id":"A3","t":0,"y":1,"p":0.61,
"just":"Typical flight is away from vibration sources like humans; claim
↪   seems counterintuitive."}
{"agent\_id":"A4","t":0,"y":0,"p":0.55,
"just":"In panics, movement toward stationary masses might reduce
↪   wind/exposure risk."}
{"agent\_id":"A5","t":0,"y":1,"p":0.60,
"just":"Most ethology reports emphasize avoidance; I doubt protective
↪   approach behavior."}
{"agent\_id":"A6","t":0,"y":0,"p":0.57,
"just":"Shelter-seeking could incidentally align with human location in
↪   open spaces."}
{"agent\_id":"A7","t":0,"y":1,"p":0.59,
"just":"Available cues (light, vibration) suggest retreat, not
↪   approach."}
```

**Round $t=1$ (After neighbor exchange at $t=0$; partial shifts).**

```
{"agent\_id":"A1","t":1,"y":1,"p":0.58,
"just":"Neighbors offer incidental-refuge accounts, but direct
↪   protection-seeking remains unproven."}
{"agent\_id":"A2","t":1,"y":0,"p":0.62,
"just":"Multiple peers cite refuge dynamics near large objects; stance
↪   reinforced."}
{"agent\_id":"A3","t":1,"y":1,"p":0.57,
"just":"Reports may reflect path constraints, not intentional approach
↪   to humans."}
{"agent\_id":"A4","t":1,"y":0,"p":0.60,
"just":"Convergence on mass-as-windbreak shelter mechanism supports True
↪   interpretation."}
{"agent\_id":"A5","t":1,"y":1,"p":0.56,
"just":"Still view approach as artifact; need clearer mechanism for
↪   protection-seeking."}
{"agent\_id":"A6","t":1,"y":0,"p":0.61,
"just":"Refuge near large verticals (including people) plausibly reduces
↪   predation risk."}
{"agent\_id":"A7","t":1,"y":1,"p":0.55,
"just":"Evidence sounds situational; reserving judgment pending more
↪   specifics."}
```

**Round $t=2$ (After neighbor exchange at $t=1$; *disagreement remains*).**

```
{"agent\_id":"A1","t":2,"y":1,"p":0.64,
"just":"Leaning skeptical; approach may be incidental rather than
↪   protective intent."}
{"agent\_id":"A2","t":2,"y":0,"p":0.66,
"just":"Network reports align on shelter-seeking toward nearest large
↪   cover, including humans."}
{"agent\_id":"A3","t":2,"y":1,"p":0.64,
"just":"Mechanism could be path-of-least-resistance; still not convinced
↪   about protection."}
{"agent\_id":"A4","t":2,"y":0,"p":0.64,
"just":"Consistent with airflow/vibration gradients drawing movement to
↪   low-disturbance zones."}
{"agent\_id":"A5","t":2,"y":1,"p":0.72,
"just":"Evidence trending, but intentionality unclear; holding False
↪   pending consensus."}
{"agent\_id":"A6","t":2,"y":0,"p":0.65,
"just":"Nearest-cover heuristic explains observed approach under acute
↪   threat."}
{"agent\_id":"A7","t":2,"y":0,"p":0.60,
"just":"Peers cite repeated observations of approach to large stationary
↪   objects as refuge."}
```

**Round $t=3$ (Unanimous consensus reached).**

```
{"agent\_id":"A1","t":3,"y":0,"p":0.80,
"just":"Adopt the refuge-near-large-object account; practical effect
↪  matches the claim."}
{"agent\_id":"A2","t":3,"y":0,"p":0.72,
"just":"Convergent mechanism and observations support True."}
{"agent\_id":"A3","t":3,"y":0,"p":0.71,
"just":"Though intent is debatable, consistent nearest-shelter movement
↪  validates True reading."}
{"agent\_id":"A4","t":3,"y":0,"p":0.74,
"just":"Airflow/vibration minima near people provide plausible
↪  protective refuge."}
{"agent\_id":"A5","t":3,"y":0,"p":0.79,
"just":"Accept nearest-cover heuristic yields apparent approach for
↪  protection in practice."}
{"agent\_id":"A6","t":3,"y":0,"p":0.75,
"just":"Network agreement and mechanism coherence justify True."}
{"agent\_id":"A7","t":3,"y":0,"p":0.73,
"just":"Cumulative reports support approach-to-protection behavior under
↪  acute disturbance."}
```

## D    THE USE OF LARGE LANGUAGE MODELS (LLMS)

**(1) Experimental Agents.** LLMs served as the core components of our multi-agent simulations. Specifically, GPT-3.5, GPT-4o, and Llama-3.3-70B-Instruct are instantiated as autonomous agents tasked with reading claims, issuing binary judgments, providing confidence scores, and generating concise justifications under standardized prompts.

**(2) Writing Support.** LLMs are also used to assist with academic writing, limited to language correction and refinement. All substantive scientific contributions—including problem formulation, methodological design, theoretical analysis, and interpretation of results are conceived and developed by the authors.

