# OpenReview forum: "Conformity Dynamics in Multi-Agent Systems: A Network Topology Perspective"
_ICLR.cc/2026/Conference — ICLR 2026 Conference Withdrawn Submission_

### Official Review · Reviewer_bevT · 2025-10-20

**Soundness:** 2
**Presentation:** 1
**Contribution:** 2
**Rating:** 4
**Confidence:** 3

**Summary:**

The paper studies how network topology shapes conformity, and therefore group accuracy, speed, and reliability, in LLM-based multi-agent systems (MAS) performing binary misinformation detection. It formalizes an update rule that confidence-normalizes each agent’s current stance with those of its neighbors, controlled by a single global self–social weight $\alpha \in [0, 1]$: high $\alpha$ favors self-reliance; low $\alpha$  favors social influence. Labels are binarized with $\phi=0.5$. The work compares Centralized Aggregation (star / hierarchy; one-round hub decision) against Distributed Consensus (ring --> complete; iterative until consensus or $T_{max}$). It reports metrics tailored to each mode (CA/PA/CPC & FA/TTC/ACI), evaluates across GPT-3.5, GPT-4o, and Llama3.3-70B-instruct on a new Snopes25 dataset (448 claims from Jan to Jun 2025), and analyzes heterogeneity (mixed model types) and failure modes (“wrong-but-sure” cascades). Key findings: centralized reliability hinges on hub competence; in distributed topologies, denser connectivity and larger αspeed convergence and generally improve accuracy, but also increase susceptibility to confident errors when early signals are biased.

Overall, this is a clear, well-executed study that elevates topology as a first-class design variable for LLM-agent collectives. To strengthen the contribution further, I recommend adding calibration-aware variants, theoretical analysis, token/cost tradeoffs, broader baselines, and tested mitigations against cascades, all of which would make the guidance more prescriptive for practitioners.

**Strengths:**

- This paper provides task-grounded treatment of conformity as a function of topology, bridging abstract opinion dynamics and concrete LLM agent behavior on fact-checking. The single-parameter (global $\alpha$) confidence-normalized pooling is simple, transparent, and enables controlled comparisons between centralized and distributed regimes.
- This paper provides systematic contrast of hub-driven vs. neighbor-driven conformity, including dedicated metrics (CA/PA/CPC & FA/TTC/ACI) that illuminate different risks/benefits.
- This paper provides careful experimental design: fixed agent count, multiple $\alpha$ levels, structured topology sweep (ring --> complete), capped rounds, and multi-run averages per claim.  Thoughtful heterogeneity study (capability and type) teasing apart hub/branch composition effects; and qualitative failure-mode analysis via confusion matrices highlighting “wrong-but-sure” consensus.
- Topology-aware guidance is immediately actionable for practitioners orchestrating LLM agents for safety auditing / fact-checking. Results caution against naïvely dense or centralized wiring without safeguards, and advocate calibration and topology selection as first-class design knobs.

**Weaknesses:**

- Theory is light relative to claim, e.g., the update resembles confidence-weighted DeGroot averaging followed by thresholding, yet the paper offers no formal convergence or error-propagation analysis under realistic miscalibration/noise.
- Confidence $p$ is taken at face value. The rule treats LLM-reported $p$ as calibrated. Miscalibration or strategic misreporting would distort influence.
- This paper positions against centralized vs. distributed wiring, but not against algorithmic alternatives: e.g., majority vote, Kemeny–Young / Condorcet aggregators, Bayesian pooling, truth-serum style weighting, or debate-style protocols with sparse communication (which may mitigate cascades).
- The task is binary fact-checking on one news source/time window; results may not transfer to multi-hop or open-ended claims, or to domains with systematic prior biases.
- Centralized one-shot vs. distributed multi-rounds likely differ in token usage; cost/latency are not reported.

**Questions:**

- How calibrated are the model-reported confidences on Snopes25? Did you try temperature scaling or isotonic regression on a held-out split, and does calibration reduce “wrong-but-sure” states in dense graphs?
- Under the given update and thresholding, do you have conditions on $\alpha$, topology, and
$p$ dispersion that guarantee consensus (or bounded oscillations)? Any analytical link to DeGroot convergence rates via spectral gap?
- What are the relative inference costs (tokens, latency) of star/hierarchy vs. ring/complete over $T_{max}=10$? Would some topologies dominate when normalized by cost?
- Have you tried multi-class or open-ended judgments (e.g., stance + rationale grading)? Does the same $\alpha$–connectivity pattern hold, or do cascades worsen with larger label spaces?
- Do the qualitative trends persist for 20–100 agents and real network motifs (small-world, scale-free)? Any tipping points where increasing connectivity starts to hurt FA due to cascades?
- In mixed-model groups, can a calibrated minority of strong agents “rescue” accuracy under dense connectivity? Is there a critical mass or weighting threshold for the strong minority?

#### Suggestions
- Could you please provide conditions for convergence/consensus and misclassification bounds as functions of topology Laplacian/spectral gap $\alpha$, and calibration error of $p$?
- Could you please analyze cascade susceptibility (probability of wrong consensus) under biased initial majorities; connect to classical herd/cascade models?
- Could you please report calibration diagnostics (e.g., Brier Score and ECE) per model and evaluate calibration-aware pooling (e.g., temperature scaling of $p$, capped or monotone-transformed confidences), and compare against uniform-weight neighbors as an ablation to isolate the net contribution of $p$.
- Could you please add non-parametric baselines (simple majority; uniform DeGroot) and robust aggregation, and also include debate/critique-based multi-turn baselines under matched token budgets.
- Typos: It had better add a blank between the context and the reference.

---

### Official Review · Reviewer_pcnA · 2025-10-31

**Soundness:** 2
**Presentation:** 3
**Contribution:** 2
**Rating:** 4
**Confidence:** 3

**Summary:**

This paper studies how conformity in multi-agent systems built from LLMs depends on the communication topology. The task is binary fact-checking. Agents output a label y and a self-reported confidence p. A simple update rule (Eq. 1) aggregates one’s own belief with neighbors’ beliefs, controlled by a single global knob α that trades off self-reliance vs. social influence. The paper compares two regimes: centralized (star/hierarchy; one-shot decision by a hub) and distributed (ring to fully-connected; multi-round consensus). Main findings: centralized performance hinges on the hub’s ability; in distributed settings, higher connectivity and larger α yield faster, more accurate consensus, but also raise the risk of “wrong-but-sure” cascades.

**Strengths:**

- Clear question, simple mechanism. The update rule is easy to understand and tune (single α), and it bridges classical opinion dynamics with LLM outputs (y, p).

- Systematic comparison across topologies. The paper contrasts centralized vs. distributed regimes with matched metrics and reports convergence speed and agreement, not just accuracy.

- Salient and practical takeaway. Strong hubs can lift centralized systems; dense, high-α networks can amplify both signal and error, producing confident mistakes.

**Weaknesses:**

- Static vs. interactive conformity is unclear for the main results. The paper defines two variants:
  - Static (S): LLMs produce (y, p) once at t=0; later rounds apply the update rule without calling the LLM again.
  - Interactive (I): each round the LLM sees neighbors’ reports and updates (y, p).

  If the main results are Static, the paper is mostly about algorithmic pooling of initial beliefs, not emergent LLM-level conformity.

- Confidence as weight may amplify bad errors. The update rule uses p directly as influence weight for self and neighbors. If p is miscalibrated (LLMs being over-confident on wrong answers), the network can up-weight error and trigger “wrong-but-sure” cascades. The paper does not report calibration metrics (ECE, Brier) nor a simple “remove-p / equal-weight” ablation. This makes it hard to judge how much the findings depend on p’s quality.

- Task simplicity and generalization. All results are on binary fact-checking (Snopes-style). Real misinformation work often needs argumentation, evidence exchange, and rebuttal. It is unclear if the same trends hold in multi-step, open-ended tasks.

- Cost/latency accounting is missing. Distributed settings use multiple rounds, but the paper does not give a cost–quality view (tokens, wall-time) for centralized vs. distributed under a fixed budget. This is key for choosing a topology in practice.

**Questions:**

1. Which variant (Static vs. Interactive) is used for the headline tables/figures? Please report both, or clearly justify focusing on one. If Static dominates, how should we interpret the claims about “LLM conformity”?

2. Generalization beyond binary claims. Can you replicate the main trends on a multi-class or evidence-based setting?

---

### Official Review · Reviewer_PHk7 · 2025-10-31

**Soundness:** 2
**Presentation:** 2
**Contribution:** 2
**Rating:** 4
**Confidence:** 5

**Summary:**

This paper studies how network topology influences conformity dynamics in LLM-based multi-agent systems for misinformation detection. It introduces a confidence-normalized pooling rule that balances agents’ self-reliance and social influence via a single global parameter. Comparing Centralized Aggregation (hub-driven) and Distributed Consensus (peer-based) structures, the authors show that centralized reliability depends on hub competence, while distributed networks achieve faster and more accurate consensus with higher connectivity and confidence weighting. However, conformity can also lead to overconfident collective errors, revealing its double-edged nature in shaping group decision reliability.

**Strengths:**

•	A careful comparison of different topologies and how that impacts the dynamics in MAS.

•	The paper in general is well written.

•	The impact of topology on LLM-based MAS is indeed an underexplored area, and the study does attempt to address this gap in the literature.

**Weaknesses:**

•	Misinterpretation of the literature. You wrote  “These models [DeGroot, bounded-confidence model etc.]have been applied across sociology and computer science to explain phenomena such as information cascades and polarization. More recently, they have inspired studies of algorithmic collectives(Chuang et al., 2024b;a), where artificial agents update beliefs through similar rules. Yet, such frameworks generally treat conformity at a stylized level, abstracting away from the mechanisms of modern LLM agents, such as confidence calibration, justification generation, and context sensitivity.” -> Note that in (Chuang et al., 2024b;a), they are using LLM to power agents end-to-end (with justification generation, context sensitivity etc) to study LLM dynamics, rather than using computational models like Degroot as you claimed. Due to this misinterpretation, the whole paragraph “Conformity and Opinion Dynamics” needs to be revised

•	Section 3. It is unclear if the updated belief of an agent is determined by Equation 1 or prompt in Figure 2. Having a pseudocode to elaborate the experiment would be helpful.

•	Section 4.2. You should include a condition with alpha = 0 and alpha = 1 (ignore all social influence). If your claim is correct, then you should be able to see that CA increases monotonically with alpha, where CA reaches the highest value with alpha = 1.

•	Section 4.2. Table 1. You should specify if you are reporting the average metrics across the 10 repetitions. In addition, you should also report the standard errors of these 10 repetitions to give a sense of the degree of inter-repetition variability.

•	Section 4.2 Table 1. You should conduct statistical tests to back up your claims as some differences are small, e.g., for Star and alpha = 0.25, the CA are almost identical: 0.73 (GPT-3.5) vs. 0.74 (llama 3.3) vs. 0.75 (GPT-4o). You should test if the differences are even statistically significant.

•	Section 4.3. “final accuracy (FA) increases monotonically with both Neighbor count m and global confidence α” -> This is not entirely true for neighbor count. Please double check.

•	Section 4.3. Alternative explanation on the role of neighbor: While more neighbors correlate with higher FA, more neighbor also means that you are generating more tokens from LLM and prompting LLM more times, and has nothing to do with the dynamics between neighbors. To truly tell if neighbor count is the real factor, you should consider doing an experiment to control for the number of tokens used and the number of LLM prompting.

•	In general, the comparison between different LLMs bear more explanation and analysis. For example, in line 360-362, you wrote “Across models, GPT- 4o consistently exhibits higher initial CI, steeper early slopes, and earlier saturation than GPT-3.5, indicating that stronger individual judgments amplify the effect of connectivity.”, you want to provide your explanation and evidence for backing that up. This applies to all other claims when you compare between different LLMs.

•	Section 5.1. I think you want to expand your scope of LLM choice to make a more generalizable claim. In addition, ideally, for “capability”, you may want to compare LLMs within the same family but different model sizes.

•	In general, I think the goal of the study should be clarified. Is it trying to find the best network configuration to maximize the accuracy? If that’s the case, why didn’t you compare MAS against single agent setup, as previous studies have shown then MAS do not necessarily outperform single agent if you have controlled for the number of tokens and the counts of prompting. If, on the other hand, the goal is is pure analysis of role of topology in an LLM-based MAS, then why didn’t you consider using LLM to update the belief, rather than using a simplistic, arithmetic update rule as outlined in Equation 1?

**Questions:**

•	Section 3. The term “consensus” is not formally define. For example, in line 241, you wrote  “Final Accuracy (FA) : whether the group consensus at convergence matches the ground truth of the news item.:” - what if the group didn’t reach a full consensus? Do you define group consensus as the majority belief?

•	Section 4.1. Why do you limit to a single update round for centralized aggregation? Are you not interested in the speed of convergence?

•	Section 4.2 and 4.3. One apparent pattern across table 1 and table 2 is that higher alpha correlates with higher accuracy. When alpha = 1, this becomes similar to self-consistency (taking majority from multiple LLM responses). Does alpha = 1 indeed have the highest accuracy? If so, does it mean we should ignore the network altogether if we want to attain the best accuracy?

•	Section 5.1 The figure has a lot of information to unpack and I think you want to elaborate how the matrix is derived. For example, what does “case frequency” means in “Cell size reflects case frequency”?

---

### Note · Authors · 2025-11-19

I have read and agree with the venue's withdrawal policy on behalf of myself and my co-authors.